# Alginate Microsponges as a Scaffold for Delivery of a Therapeutic Peptide against Rheumatoid Arthritis

**DOI:** 10.3390/nano13192709

**Published:** 2023-10-05

**Authors:** Daniela Ariaudo, Francesca Cavalieri, Antonio Rinaldi, Ana Aguilera, Matilde Lopez, Hilda Garay Perez, Ariel Felipe, Maria del Carmen Dominguez, Odalys Ruiz, Gillian Martinez, Mariano Venanzi

**Affiliations:** 1Department of Chemical Science and Technologies, University of Rome Tor Vergata, Via della Ricerca Scientifica 1, 00133 Rome, Italy; daniela.ariaudo@alumni.uniroma2.eu (D.A.); francesca.cavalieri@uniroma2.it (F.C.); 2PROMAS-MATPRO Laboratory, Sustainability Department, ENEA, 00123 Rome, Italy; antonio.rinaldi@enea.it; 3NANOFABER S.r.l., Via Anguillarese 301, 00123 Rome, Italy; 4Biotechnological Development Direction, Center for Genetic Engineering and Biotechnology, Havana 10600, Cuba; ana.aguilera@cigb.edu.cu (A.A.); matilde.lopez@cigb.edu.cu (M.L.); odalys.ruiz@cigb.edu.cu (O.R.); 5Biomedical Research Direction, Center for Genetic Engineering and Biotechnology, Havana 10600, Cuba; hilda.garay@cigb.edu.cu (H.G.P.); ariel.felipe@cigb.edu.cu (A.F.); mcarmen.dominguez@cigb.edu.cu (M.d.C.D.); 6Business Development Direction, Center for Genetic Engineering and Biotechnology, Avenue 31/158 and 190, Playa, Havana 11600, Cuba; gillian.martinez@cigb.edu.cu

**Keywords:** alginate, CIGB814, drug delivery, polysaccharide microsponges, rheumatoid arthritis, therapeutic peptides

## Abstract

The quest for biocompatible drug-delivery devices that could be able to open new administration routes is at the frontier of biomedical research. In this contribution, porous polysaccharide-based microsponges based on crosslinked alginate polymers were developed and characterized by optical spectroscopy and nanoscopic microscopy techniques. We show that macropores with a size distribution ranging from 50 to 120 nm enabled efficient loading and delivery of a therapeutic peptide (CIGB814), presently under a phase 3 clinical trial for the treatment of rheumatoid arthritis. Alginate microsponges showed 80% loading capacity and sustained peptide release over a few hours through a diffusional mechanism favored by partial erosion of the polymer scaffold. The edible and biocompatible nature of alginate polymers open promising perspectives for developing a new generation of polysaccharide-based carriers for the controlled delivery of peptide drugs, exploiting alternative routes with respect to intravenous administration.

## 1. Introduction

The efficacy of a drug compound critically depends on the administration method. Optimal drug administration should minimize degradation by enzymatic attack, avoid harmful side effects, and enhance bioavailability in the target area [1]. Commonly employed delivery methods require a suitable formulation of the drug, like tablets or solutions for oral or intravenous administration [2]. However, these traditional methods may show severe drawbacks, like drug overdose, toxic effects, and inefficient pharmacokinetics [3,4].

New strategies for delivery of drugs rely on their encapsulation into polymeric materials that allow a continuous drug release by slow diffusion through the polymer matrix, or controlled degradation of the same matrix [5]. In the last decade, several nanostructured biocompatible drug carriers have been developed, i.e., polymer microparticles and microcapsules, lipoproteins, liposomes, and micelles [6]. An alternative carrier is represented by polymeric microsponges (PMS), globular aggregates characterized by a continuous network of nanochannels that form structured microcavities [7]. PMS have found applications in biomedicine for topical therapeutic treatments, but they have recently been taken under consideration for the oral administration of drugs using edible polymers [8].

PMS technology provides several advantages to drug delivery: a clinically safe release, prolonged and programmable administration times, easy and low-cost production, enhanced thermal stability, and enhanced physico-chemical stability. PMS have been shown to adsorb and stably embed a huge number of bioactive compounds, and are easily incorporated into gels, creams, liquids, or powder substrates [9]. The outer surface of PMS is highly porous, and their inner structure is rich of microcavities, where drug molecules can be stably embedded in a protective environment. Suitable engineering of PMS allows control of the inner/outer fluidics of the bioactive compounds and—through careful modulation of the polymer/drug interaction within the polymer microcavities—the drug time release to the target. A prolonged time release will permit lower dosages, less frequent drug administration, and therefore, lower toxicity and reduced probability of allergic reactions [10].

PMS have shown to be not irritating, not mutagenic, not toxic, and not allergenic. In addition to that, PMS can be designed so that the average pore size can be less than 0.25 μm, inhibiting bacterial permeation through their inner structure [11].

Specific advantages of PMS are as follows: (i) the large and porous outer surface. In particular, the elasticity of the polymeric matrix can be modulated by varying the nature and composition of the polymer and the degree of cross-linking; (ii) the method of synthesis is simple and not expensive; (iii) the flexible design and engineering. The PMS size and porosity determine the loading capacity of PMS and their release properties in a specific environment [12] and under external stimuli; i.e., temperature, pressure, or pH jumps [13].

We recently reported on the synthesis of polysaccharide-based microsponges as a carrier for proteins and small molecules [14]. Herein, we report on the synthesis and characterization of macroporous PMS formed by sodium alginate (NaAlg) using a di(N-imidazolyl)carbonylaminoethyl disulphide cross-linker [15]. Alginate is a natural, non-toxic, and inexpensive anionic polymer extracted from brown algae in an alkaline solution, and it is formed by linear polysaccharides with variable fractions of β-D-mannuronic (M) and α-L-guluronic (G) units connected by 1,4′ linkages [16,17]. Unlike other types of microsponges, which are prepared using emulsion polymerization, our polysaccharide PMS are prepared by a co-precipitation method. In this approach, the polysaccharide chains in solution mediate the nucleation and growth of cross-linking agent crystals, leading to the formation of a macroporous crosslinked structure with redox responsive behaviour [15]. The porosity of the as-prepared PMS can be attributed to two possible events: (i) the reaction of the cross-linker with carboxylic acid moieties produces CO_2_, which likely acts as a porogenic agent and leads to the formation of macropores within the polymer matrix; (ii) the polysaccharide chains promote the nucleation of cross-linker crystals that serve as a “glue” to hold polymer agglomerates together.

The morphology of NaAlg PMS was characterized by optical, confocal fluorescence and scanning electron microscopy experiments. Furthermore, their efficiency in the loading and release of a therapeutic peptide, synthesized at the Center for Genetic Engineering and Biotechnology (CIGB) in Havana (Cuba), and denoted as CIGB814, was investigated using spectroscopy methods. CIGB814 is an altered peptide ligand derived from the human heat shock protein 60 (HSP60), a self-antigen involved in the pathogenesis of rheumatoid arthritis (RA). It consists of a 27-residue long sequence (SIDLKDKYKNIGAKLVQLVANNTNEEA-NH_2_), and it is presently under a phase 3 clinical investigation for RA therapy. CIGB814 exerts its therapeutic effect inhibiting the activation of T cells and stimulating the production of regulatory cells that are able to suppress the activity of CD4+ cells [18]. In the phase I clinical trial with RA patients, CIGB814 has been shown to reduce the levels of tumor necrosis factor (TNFα) and interleukin -17, and it also induced a significant decrease in auto-antibodies against citrullinated self-proteins [19]. Interestingly, this peptide has also been used in the treatment of COVID-19, with promising results. Of note, the aggregation properties of CIGB814 have recently been characterized in our laboratory using spectroscopic techniques and molecular dynamics simulation [20]. We found that the CIGB814 aggregation pathway is a multistep process strongly dependent on the peptide concentration. At micromolar concentrations, small peptide oligomers formed compact structures stabilized by interactions between hydrophobic residues, that at a concentration higher than 100 μM, gave rise to peptide globules, as typical of an aggregation process driven by a hydrophobic effect. Finally, at a concentration greater than 500 μM, the coalescence of those globular structures led to the formation of peptide fibrils of micrometric length. This hierarchical self-assembly process is typical of the aggregation modes of bioactive peptides, allowing fine-tuning of the design of nano- and mesoscopic peptide architectures [21]. In this context, the comprehension of the aggregation properties of CIGB814 is important to control the monomer/aggregate equilibria in both the nanostructured environment (peptide loading experiment) and the aqueous solution (peptide release experiment).

## 2. Materials and Methods

### 2.1. Materials

Di(imidazol-1-yl)methanone (carbonyl di-imidazole, CDI), 2-(2-aminoethyldisulfanyl)ethanamine (Cys), sodium alginate (NaAlg), [9-(2-carboxy-6-isothiocyanatophenyl)-6-(diethylamino)xanthen-3-ylidene]-diethylazanium-chloride (Rhodamine B isothiocyanate, RBITC), and dipotassium-trisodium-dihydrogen phosphate-hydrogen phosphate-dichloride (1:1:1:1) (phosphate buffered saline, PBS) were purchased from Merck Italia (Milan, Italy).

*PMS synthesis.* The synthesis of cross-linked NaAlg PMS was carried out using di(N-imidazolyl)carbonylaminoethyl disulphide (DIDS) as a cross-linker (Figure 1). The latter was obtained by reacting 30 mg of CDI and 20 mg of Cys. In particular, Cys was dissolved in 300 μL MilliQ water and vortexed for 30 s. Subsequently, CDI was added in one shot, and precipitated crystals formed immediately. The precipitate was stirred in vortex for 1 min, and the mixture was incubated for 30 min at room temperature [15]. After that, 50 µL of HCl 5M were added to the precipitate until complete dissolution (final pH comprised between 2 and 4).

NaAlg PMS were produced by adding 1 mL of 1% (*w*/*v*) NaAlg to the cross-linker (36 mg/mL, 3.6% (*w*/*v*)). After 72 h shaking (T-Shaker, 37.5 °C, 250 rpm), a white precipitate was obtained, washed 3 times with Milli-Q water, and centrifugated each time for 5 min at 5000 RPM to eliminate unreacted cross-linker and polymer molecules. The amount of polysaccharide forming the synthesized PMS was determined by functionalizing the alginate chain with RBITC. To this aim, a UV-Vis calibration curve was determined, measuring the absorbance at λ = 545 nm of RBITC solutions of different concentrations (from 0.003 to 0.2 mg/mL). Specifically, 1 mg of RBITC was added to a 2% (0.02 g/mL) NaAlg aqueous solution, and the unreacted probe was washed away by a Medicell International LTD dialysis membrane (cut-off = 12–14,000 Da). The yield of RBITC-NaAlg labelling and the amount of RBITC-NaAlg embedded in the PMS were determined by measuring the RBITC absorbance at λ = 545 nm in the supernatant solution and subtracting this value from that one of the initially added fluorophore. A scheme of the reaction and the UV-Vis calibration curve of RBITC are reported as Appendix A in Appendix A, respectively.

CIGB814 (SIDLKDKYKNIGAKLVQLVANNTNEEA-NH_2_, MW = 2988.38 g·mol^−1^) was synthesized at the CIGB by solid-phase synthesis using Fmoc/tBu chemistry, purified by reverse-phase liquid chromatography, and characterized using mass spectrometry, as reported elsewhere [20].

### 2.2. Experimental Methods

The NaAlg average MW (*M_v_*) was determined using viscosity measurements, correlating the intrinsic viscosity [*η*] of a NaAlg diluted solution to its *M_v_* through the Mark–Houwink–Sakurada equation:(1)$$\left[\eta \right]=kM_{v}^{a\;}\;$$
where *k* and *a* are the Mark–Houwink constants, characteristic of a specific polymer/solvent system at a given temperature. [*η*] can also be obtained by extrapolating the Huggins and Kramer equations, expressed in term of the specific ([*η_sp_*]) or relative ([*η_rel_*]) viscosity, respectively, at null polymer concentration [22]. Viscosity measurements were carried out for 0.1, 0.25, 0.5, 0.75, and 1 mg/mL NaAlg solutions in 0.1 M NaCl at T = 25 °C.

UV–Vis absorption experiments were carried out at room temperature by a Jasco V-770 spectrophotometer using quartz cells [l = 1 cm, (Hellma Italia, Milan, Italy)]. The extinction coefficient (ε_0.1%_) was determined at *λ_max_* = 280 nm with respect to a 0.1% (1 mg/mL) peptide solution.

Steady-state fluorescence spectra were measured by a Fluoromax 4 (Horiba, Kyoto, Japan) with single photon counting detection, using quartz cells (l = 1 cm) at room temperature. Fluorescence microscopy experiments were carried out on an Axio Scope.A1 (Zeiss, Oberchoken, Germany) microscope, equipped with an HBO 50 Hg lamp and an AxioCam ICm1 CCD camera.

Confocal laser scanning fluorescence microscopy (CLSFM) experiments were performed using an FV1000 (Olympus, Tokyo, Japan) confocal scanning device interfaced to an Olympus IX-81 inverted microscope. CLSFM measurements were carried out using an oil-immersion 60x objective (numerical aperture: 1.42). Three-dimensional images of the samples investigated were reconstructed using the Imaris 6.2.1 (Bitplane, Belfast, UK) software.

Scanning electron microscopy experiments were carried out using a field-emission environmental scanning electron microscope (FE-ESEM) LEO 1530 (Zeiss), after deposition of a 5 µL PMS solution on HOPG under UHV conditions.

Dynamic light scattering experiments were performed at room temperature by a DLS Zetasizer Nano ZS (Malvern Instruments, Malvern, UK), using a 1 cm quartz cuvette. The hydrodynamic diameter (*d_H_*) of a microparticle was associated to its translational diffusion coefficient (*D*) through the Stokes–Einstein equation:(2)$$d_{H}=\frac{k_{B}T}{3\pi \eta D}$$

Peptide loading to NaAlg PMS in an aqueous solution was studied at a 0.5 mg/mL CIGB814 concentration, and 4 and 8 mg PMS; i.e., for 1:8 and 1:16 (*w*/*w*) peptide/PMS ratios. NaAlg PMS were preliminarily dehydrated, and then added to the peptide aqueous solution to enable the loading of peptide into the polymer matrix by swelling. The samples were incubated at room temperature on a rotary shaker. The amount of loaded peptide was determined by analysing the supernatant solution using UV–Vis and fluorescence spectroscopy after centrifugation (2000 rpm for 5 min) of the peptide/PMS aqueous solutions at different peptide incubation times (0, 0.5, 1, 2, 3 and 4 h). Specifically, the Tyr absorption at *λ_max_* = 280 nm and the fluorescence emission at *λ_max_* = 307 nm were measured for the peptide initially added to the PMS solution (I_0_), the peptide in the supernatant solution (I_t_) at different times t and the blank (NaAlg PMS solution, I_b_). The fraction of loaded peptide (x_lp_) was obtained by the equation:(3)$${\;x}_{\mathrm{lp}}=\frac{I_{0}-\left(I_{t}-I_{b}\right)}{I_{0}}100$$

Control experiments were carried out on NaAlg PMS and CIGB814 separate solutions.

Peptide release from PMS for the 1:8 and 1:16 (*w*/*w*) peptide/PMS ratios was studied at pH 7.4 in PBS to mimic physiological conditions. The peptides were incubated in PMS for 2 h, and the supernatant solutions analysed using UV–Vis absorption and fluorescence spectroscopies. After washing and drying the loaded PMS, they were incubated at 37 °C with 1 mL of PBS solution (0.15 M, pH 7.4). At different time intervals, the sample was centrifuged, 0.5 mL of the soluble supernatant was taken for analysis, and another 0.5 mL of fresh PBS was added. Supernatant solutions on dehydrated PMS were also analysed as control experiments. The kinetics of peptide fractional release (x_rp_) were analysed by measuring its time-dependent fluorescence emission intensities in the supernatant solution, considering that the peptide loaded after 2 h of PMS incubation as the starting point (t = 0) of the release experiment. Specifically, the sum of the emission intensities (F_RP,tot_) at $\lambda_{max}$ = 307 nm measured at each incubation time t from 0 to 4 h (F_t_) and subtracted by half the fluorescence intensity of the previous step (F_(t−1)/2_) (this was necessary because at each step, half of the supernatant solution was picked up to measure the fluorescence intensity of the released peptide) was divided by the emission intensity at the same wavelength of the peptide loaded after a 2 h incubation time (F_0_) (Equation (4)):(4)$${\;x}_{\mathrm{rp}}=\frac{F_{\mathrm{RP},\mathrm{tot}}}{F_{0}}100=\frac{\sum_{t}\left(F_{t}-F_{\left(t-1\right)/2}\right)}{F_{0}}100$$

### 2.3. Statistical Methods: DoE Factorial Design Approach for the Estimate of PMS Diameter

The diameter size distribution of NaAlg PMS is strongly dependent on the effect of process parameters. Experimental methods from Design of Experiment (DoE) provide a statistical approach to investigate and rationalize this aspect [23,24]. In particular, by considering the average PMS diameter (µm) provided by a DLS experiment as the output parameter *Y* of a formulation, we focus on the effects of Cys and CDI weights (mg), and H_2_O volumes (µL) taken as factors or regressors. Output and input variables are supposed to be linked by a functional relation
(5)$$Y\left(diameter\right)=f\left(Cys,CDI,H_{2}O\right)$$
which is here studied by a linear regression approach, named 2^3^ full factorial design of DoE, to assess a model of general form:(6)$$\left(diameter\right)=C_{0}+C_{Cys}Cys+C_{CDI}CDI+C_{H_{2}O}*H_{2}O+C_{CysCDI}Cys*CDI+C_{CysH_{2}O}Cys*H_{2}O+C_{CDIH_{2}O}CDI*H_{2}O+C_{CysCDIH_{2}O}Cys*CDI*H_{2}O$$

Equation (6) captures the effect of the three main parameters “Cys, CDI and H_2_O” (main effects) and their combinations “Cys*CDI, Cys*H_2_O, CDI*H_2_O, Cys*CDI*H_2_O” (first order interactions) on the PMS diameter. The three main parameters are varied as in Table 1, landing eight independent formulation variants (runs) for all possible combinations of X_1_, X_2_, and X_3_, which were performed in duplicates. Upon measuring the average PMS diameters for each run, a dataset of 16 statistical observations suitable for the ANOVA analysis was obtained and analysed, as reported in detail in the Appendix A. Through the statistical software MINITAB 16© (Minitab, PA, USA), a backward elimination search algorithm was used to obtain a reduced model with the most significant parameters (the statistical significance of each variable being measured by *p*-value), ranked for importance in descending order by increasing *p*-values, and retained in the model for *p*-values lower than a significance threshold, in this study taken as 0.25.

## 3. Results and Discussion

### 3.1. Characterization of Alginate Microsponges

The structural properties of alginate PMS are determined by the network of intra- and intermolecular cross-linkages formed by DIDS with the NaAlg polysaccharide. In particular, the PMS porosity is determined by the reaction between the cross-linker molecules and the hydroxyl and carboxyl groups of Guluronate. Clusters formed by cross-linked polysaccharides eventually nucleate the growth of porous microsponges [15].

The NaAlg average MW [MW(NaAlg) = 78.6(±0.1) kDa] was determined by viscosity measurements through the Mark–Houwink–Sakurada equation. Experimental reduced and inherent viscosities at different polysaccharide concentrations are reported in the Appendix A.

NaAlg was functionalized with RBITC to determine the amount of polysaccharides forming the synthesized PMS. RBITC-labelled NaAlg with a substitution degree of 23% was used in the preparation of PMS, and the resultant supernatant was analysed by UV–Vis absorption at λ = 545 nm to quantify the content of NaAlg embedded in the PMS (70%). The emission spectra of the RBITC/NaAlg in aqueous and in the supernatant solutions were reported as Appendix A. Fluorescence intensities were normalised by the absorbance at the excitation wavelength (λ = 545 nm) and corrected for the inner filter effect. The high fluorescence intensity shown by RBITC/NaAlg in the supernatant solution, together with the observed turbidity of the solution, suggests that some RBITC/NaAlg PMS are still present in solution, inhibiting the quenching of the dye label by water molecules. The quenching of the RBITC fluorescence quantum yield in aqueous solutions can be observed in the RBITC-Alg concentration-dependent emission spectra reported in the Appendix A, where the more concentrated polymer solutions showed definitely less intense RIBTC fluorescence emission spectra.

Optical microscopy measurements showed that NaAlg PMS attain a globular morphology, characterized by a dense inner phase coated by an irregularly jagged layer, and diameters roughly between 0.6 and 8 µm (Figure 2A). Interestingly, cross-linker molecules under the same experimental conditions were shown to form only irregular micrometric conglomerates (Appendix A).

A more detailed inspection of the NaAlg PMS morphology can be obtained using SEM experiments with nanometric resolution. In the SEM image reported in Figure 2B, the porous structure of the polysaccharide PMS is clearly visible, showing porous globular structures of regular dimensions with diameters of around 3–4 μm, and pore sizes between 50 and 120 nm. For comparison, the microstructures formed by the sole cross-linker molecules and the NaAlg polymer were reported in Appendix A, confirming the role of the polysaccharide matrix in the formation of regularly shaped PMS.

The globular morphology of NaAlg PMS, the alginate chains of which were functionalized with RBITC fluorophores in a 10:1 polymer/fluorophore ratio, was clearly shown by the CLSFM images reported in Figure 3. The 2D section of RIBTC-labelled NaAlg reported in Figure 3B showed a rather homogeneous coloration, suggesting that the observed globular PMS are densely filled by the marked polysaccharide chains.

The CLSFM images were analysed through the ImageJ software, allowing us to estimate the size distribution of NaAlg/RIBTC PMS (Appendix A). The frequency histogram of the diameters was reproduced by a Gaussian function, showing an average diameter of 2.8 ± 0.9 μm and a relatively large dispersion index (31.2%).

DLS experiments were also carried out to determine the diameter size distribution of different formulations of NaAlg PMS, with the aim to optimize the PMS preparation conditions. To convey the effects of process parameters on the average diameter size of NaAlg PMS and select a numerical estimate of Equation (6), the “backward elimination” search algorithm was applied obtaining the following (reduced) model:(7)$$diameter=-10925+97.5\;H_{2}O+24.90\;Cys\cdot CDI+2.669\;Cys\cdot H_{2}O-1.232\;CDI\cdot H_{2}O$$

This model is adequate for capturing the variability in the used experimental dataset, with a fitting coefficient of determination R^2^ = 81% (Appendix A), and indicates that Cys and CDI main factors play a minor role, alongside the Cys*CDI*H_2_O cross variable (i.e., all characterized by a *p*-value larger than 0.25). Instead, the H_2_O volume (*p*-value = 0.11), CDI*H_2_O (*p*-value = 0.084), Cys*CDI (*p*-value = 0.002), and Cys*H_2_O (*p*-value = 0.002) play a more significant role, ranked in ascending order for importance. The Cys*H_2_O interaction is therefore the single mostly effective parameter in determining the average PMS diameter, followed by Cys*CDI, H_2_O, and CDI*H_2_O. As a cautionary warning, it is remarked that this model, while insightful, is intended to provide a first order description of this relationship, and it is deemed to provide an upper bound estimate of the real average PMS diameter distribution, since DLS data in this particle range is biased and overestimated primarily due to both the loss of method accuracy above ~600 nm and the outlier PMS corresponding to clusters counted erroneously as a single particle. Details of the performed DoE study are reported in the Appendix A.

Furthermore, by analyzing the results of DLS measurements carried out under different experimental conditions, it can be seen that all the experiments can be described taking into account mono- and bimodal size-diameter distributions depending on the specific composition of the preparative formulation (Appendix A). In particular, monomodal distributions showed an average diameter of 5.4(±1.8) μm, while bimodal distributions showed average size diameters of 7.1(±1.8) and 1.8(±0.7) μm. The more stable Z-average parameter, obtained by cumulant analysis of the overall DLS data [25,26], was between a minimum size of 1.78 μm and a maximum size of 8.2 μm, the average size being 5.1(±1.7) μm. Of note, the polydispersity indices of all the DoE experiments were found to be well below the confidence value of 0.7, attesting to the quality of the performed DLS measurements (Appendix A). However, it should be emphasized that the reported average values represent only an indication of the general size of the PMS due to the different formulations for which the DLS data were obtained, and were reported here only to provide a general description of the as-prepared PMS without entering in the details of the specific diameter distribution obtained for each preparative condition for brevity.

### 3.2. Peptide Loading

Peptide loading by swelling in an aqueous solution of pre-dehydrated NaAlg PMS was investigated for 0.5 mg/mL CIGB814, and 4 mg/mL and 8 mg/mL NaAlg PMS; i.e., for 1:8 (*w*/*w*) and 1:16 (*w*/*w*) peptide/NaAlg PMS ratios. It should be stressed that direct loading of CIGB814 to the PMS solution was quite inefficient, indicating a low solubility of the peptide within the PMS. Moreover, at the used concentration, 167 μM, CIGB814 is present in solution as a mixture of monomer and aggregate species. It is therefore most likely that during the swelling process, the peptide would adsorb to the PMS in both forms.

The UV absorption spectra of the supernatant solutions obtained after centrifugation of a 1:8 and 1:16 CICB814/NaAlg PMS ratio at different incubation times are reported in Figure 4A and Figure 4C, respectively. The observed absorption bands are associated with the $n\to \pi^{*}$ transition of the Tyr residue $(\lambda_{max}$ = 280 nm, $\varepsilon_{280}$ = 1490 M^−1^cm^−1^, $\varepsilon_{0.1\%}$ = 0.493). The amount of loaded peptide was estimated by subtracting the absorbance at $\lambda_{max}$= 280 nm of the supernatant solution measured at each incubation time from 0 to 4 h from that of the initially added peptide (Equation (3)). Raw data on the peptide loading (0.5 mg/mL) to 4 mg and 8 mg NaAlg PMS are reported in the Appendix A (Appendix A, respectively).

In Figure 4B, the fractional loading of CIGB814 with respect to the added peptide to PMS solutions was also reported, showing that in the case of 1:8 peptide/PMS formulation, the loading of CIGB814 by swelling requires a 4 h incubation time to obtain about 34% loading efficiency.

It should be noted (Figure 4C) that the assessment of the loaded peptide by the UV-Vis absorption data for the 1:16 peptide/PMS formulation is somewhat blurred by a marked distortion of the tyrosine absorption band caused by diffuse light contamination, indicating that fragments of the PMS are present in the supernatant solution. However, in a semi-quantitative way, the UV–Vis data suggest that, for a 1:16 CIGB814/NaAlg PMS ratio, the fraction of loaded peptide increased to about 75% after a 4 h incubation time.

The intrinsic fluorescence of the Tyr residue was also exploited to estimate the loading efficiency of CIG814 on NaAlg PMS through an independent experiment. Peptide solutions were excited at $\lambda_{ex}$ = 270 nm and the spectra recorded between 285 and 400 nm. The emission spectra of supernatant solutions of CIB814 (0.5 mg/mL) and NaAlg PMS (4 and 8 mg) at different times (from t = 0 to t = 4 h) were reported in Figure 5A and Figure 5C, respectively, together with the associated peptide fractional loading (Figure 5B,D), estimated by subtracting the emission intensity at $\lambda_{max}$ = 307 nm of the supernatant solution measured at each incubation time from 0 to 4 h from that of the initially added peptide (Equation (3)). Raw data of these experiments are reported in the Appendix A.

Fluorescence data are in close agreement with the UV–Vis absorption results. In particular, the peptide fractional loading increases from 38%—measured in the case of 4 mg/mL NaAlg PMS—to about 75% for 8 mg/mL polymer solutions. It should be emphasized that fluorescence data are less affected by light scattering contamination of the PMS or PMS fragments eventually present in the supernatant solution, and therefore fluorescence techniques can be considered the method of election for these studies [27].

### 3.3. Peptide Release

The kinetics of CIGB814 release from 4 and 8 mg/mL NaAlg PMS was studied in a PBS solution (pH 7.4), measuring the fluorescence emission of the Tyr residue at different times (0–120 h) and taking the peptide loaded after 2 h incubation time as the starting point of the release experiment (t = 0); i.e., 0.19 and 0.37 mg/mL from 1:8 and 1:16 peptide/PMS formulations, respectively.

In Figure 6, we reported the absorption spectra of supernatant solutions of NaAlg PMS in PBS (pH = 7.4) measured at different incubation times. The reported spectra are characterized by a weak absorption band that peaked at $\lambda_{max}$ = 250 nm, associated with the $n\to \pi^{*}$ transition characteristic of the disulphide (-S-S-) bond. Interestingly, for supernatant solutions extracted within the first 2 h of incubation time, the absorbance of (-S-S-) $n\to \pi^{*}$ transition can be clearly detected, while at longer times, scattered light contributions markedly affect the measured spectral features. This effect can also be observed in the UV–Vis absorption spectra of NaAlg PMS loaded with CIGB814, where the absorption of the Tyr group $(\lambda_{max}$ = 270 nm) partially overlapped the disulphide absorption band (Appendix A). The results shown in Figure 5 and Appendix A indicate that the NaAlg PMS, under the applied conditions of the peptide release experiments, are subjected to partial degradation, dissolving small PMS and/or PMS fragments in the PBS solution. This effect clearly increases the porosity of the polymer matrix and promotes the release of the loaded peptide [28].

As the absorption spectra of CIGB814/NaAlg PMS are affected by the degradation process described above, the release of CIGB814 from the PMS scaffold was characterized by fluorescence emission experiments, using Tyr as an intrinsic fluorescent probe. The cumulative fraction of released peptide was determined by measuring the Tyr fluorescence intensity at $\lambda_{max}$ = 307 nm (Figure 7) of supernatant solutions at different incubation times, as described in the experimental methods section (Equation (4)). The emission spectra of CIGB814/NaAlg PMS in PBS (pH 7.4) are reported as Appendix A.

The CIGB814 peptide molecules released during the first 2 h are reported in the insets of Figure 7A,B. Raw data of these experiments are reported as Appendix A.

From these results, it appears that the cumulative fractional release of CIGB814 is almost independent from the PMS quantity, amounting to about 72% for 4 mg NaAlg PMS, and about 68% for 8 mg NaAlg PMS. According to this evaluation, about 30% of the peptide molecules remain adsorbed on the microsponges, most likely confined to the PMS macropores.

The kinetic behaviour of peptide release from NaAlg PMS reported in the insets of Figure 7A,B indicate a fast peptide release dominated by a diffusion mechanism, favoured by partial degradation of the polysaccharide matrix. This finding strongly suggests that the majority of the peptide is physically adsorbed on the surface of the PMS cavities. Of note, the peptide confined inside the nanopores can be released upon disassembly of the PMS in a reducing environment [14]. Overall, these results indicate that NaAlg PMS provide a suitable platform to protect CIGB814 against proteolytic degradation.

### 3.4. Imaging of CIGB814/NaAlg PMS

#### 3.4.1. Fluorescence Confocal Microscopy

CIGB814/NaAlg(RIBTC) PMS were imaged by CLSFM [29]. The corresponding 3D-reconstructed confocal fluorescence images reported in Figure 8 showed that the loading of the therapeutic peptide promotes PMS aggregation, most likely because of partial neutralization of the PMS surface charge. This also implies a predominant localization of the peptide on the PMS surface, as suggested by the homogenous coating of the globular PMS shown by the CLSFM images reported in Figure 8.

#### 3.4.2. Scanning Electron Microscopy (SEM)

A SEM image of NaAlg PMS (4 mg) loaded with 0.5 mg CIGB814 (incubation time 2 h) is shown in Figure 9. It should be noted that, as the SEM experiments are carried out on dehydrated samples, the PMS dimensions provided by these measurements are typically lower than those obtained by optical or fluorescence confocal microscopies.

In agreement with the optical microscopy results that highlighted a strong perturbation of the loaded peptide on the PMS morphology, the SEM images of peptide-loaded PMS also showed extended aggregation and a marked change of the PMS morphology. In particular, the PMS outer surface presents a rosebud-like aspect, characterized by indented microstructures embedding nanometric cavities. This remarkable modification of the PMS morphology strongly supports the idea that CIGB814 molecules are mainly adsorbed on the polymer surface, as also suggested by CLSFM imaging.

## 4. Conclusions

The morphology and stability of PMS formed by sodium alginate polysaccharides cross-linked by di(N-imidazolyl)carbonylaminoethyl disulphide [15] were characterized, together with the loading and release capacities of CIGB814, a therapeutic peptide fighting RA pathogenesis, presently under a phase 3 clinical trial.

NaAlg PMS characterization, carried out by optical, electron scanning, and fluorescence confocal microscopies, revealed that alginate PMS attained a nanoporous globular structure, rather similar to the morphology of the formerly investigated hyaluronic acid microsponges [14], although the former PMS exhibited pores of reduced diameters. This property significantly affects the load efficiency and release kinetics of peptide molecules suitably embedded in the PMS microcavities.

In the case of the NaAlg PMS investigated, CIGB814 exhibited a very high loading efficiency (75% after 4 h) for a 1:16 peptide/PMS ratio. Under these experimental conditions, the PMS release efficiency amounted to 67% after 2 h, while around 30% of the peptide molecules remained entrapped in the polymeric matrix. Interestingly, the observed release kinetics is typical of a diffusion-driven process under low drug solubility and partial matrix erosion conditions [30].

These results pave the way for developing a macroporous platform based on cross-linked alginate polymers, and open promising perspectives for pursuing the oral administration of peptide drugs, using edible polysaccharides as polymer building blocks. The stability and bioavailability of the loaded peptide in the gastrointestinal environment will be explored in future studies.

## Figures and Tables

**Figure 1 nanomaterials-13-02709-f001:**
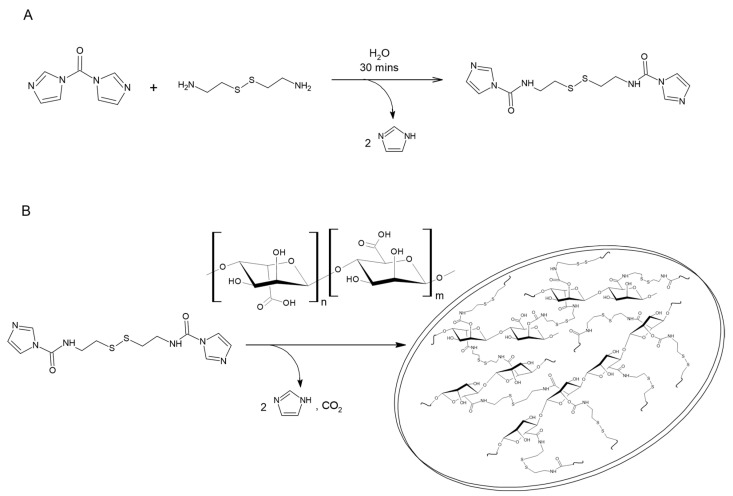
(**A**) Synthetic scheme of di(N-imidazolyl)carbonylaminoethyl disulphide (cross-linker); (**B**) Synthetic scheme of sodium alginate microsponges.

**Figure 2 nanomaterials-13-02709-f002:**
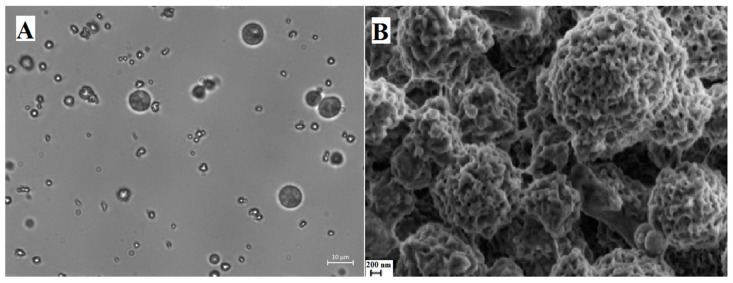
(**A**) Optical microscopy images of NaAlg PMS (60×, scale bar: 10 μm); (**B**) SEM image of NaAlg PMS (scale bar: 200 nm).

**Figure 3 nanomaterials-13-02709-f003:**
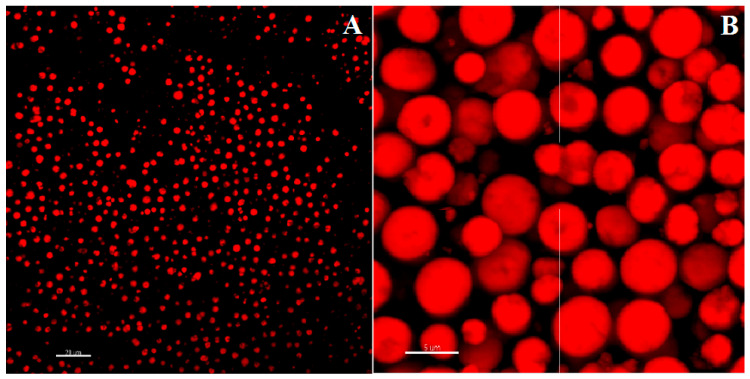
Two-dimensional fluorescence confocal microscopy images of NaAlg PMS functionalized by Rhodamine B (RBITC) in a 10:1 polymer/fluorophore ratio. (**A**) scale bar: 20 μm; (**B**) scale bar: 5 μm.

**Figure 4 nanomaterials-13-02709-f004:**
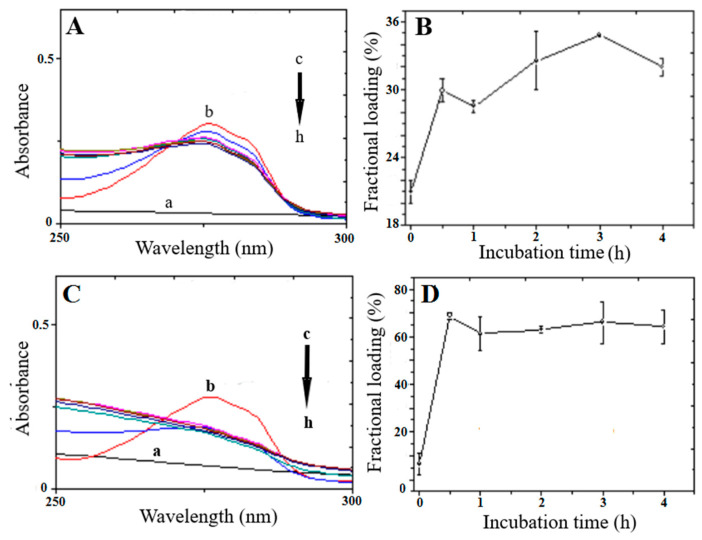
(**A**,**C**): UV–Vis absorption spectra of NaAlg PMS/peptide supernatant solutions. CIGB814: 0.5 mg/mL; NaAlg PMS: 4 mg/mL (**A**), 8 mg/mL (**C**). (a) NaAlg PMS; (b) CIGB814; (c) CIGB814/NaAlg PMS at t = 0; (d) t = 30 min; (e) t = 1 h; (f) t = 2 h; (g) t = 3 h; (h) t = 4 h. (**B**,**D**) fractional loading of CIGB814 (%) at different times (h) calculated from the UV-Vis absorption data reported in (**A**,**C**), respectively (λ_abs_ = 280 nm).

**Figure 5 nanomaterials-13-02709-f005:**
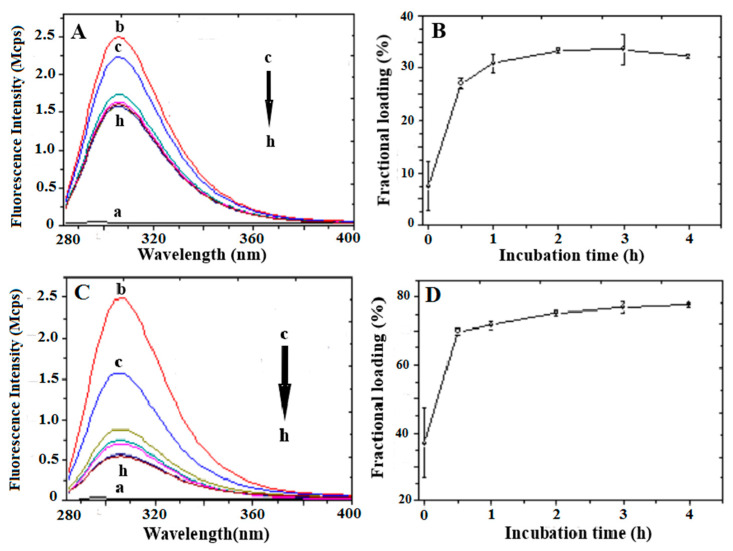
(**A**,**C**): Fluorescence emission spectra (Mcps=10^6^ count per second) of NaAlg PMS/peptide supernatant solutions. CIGB814: 0.5 mg/mL; NaAlg PMS: 4 mg/mL (**A**), 8 mg/mL (**C**). (a) NaAlg PMS; (b) CIGB814; (c) NaAlg PMS +CIGB814 at t = 0; (d) t = 30 min; (e) t = 1 h; (f) t = 2 h; (g) t = 3 h; (h) t = 4 h. (**B**,**D**) fractional loading of CIGB814 (%) at different times (h) (λ_em_ = 307 nm).

**Figure 6 nanomaterials-13-02709-f006:**
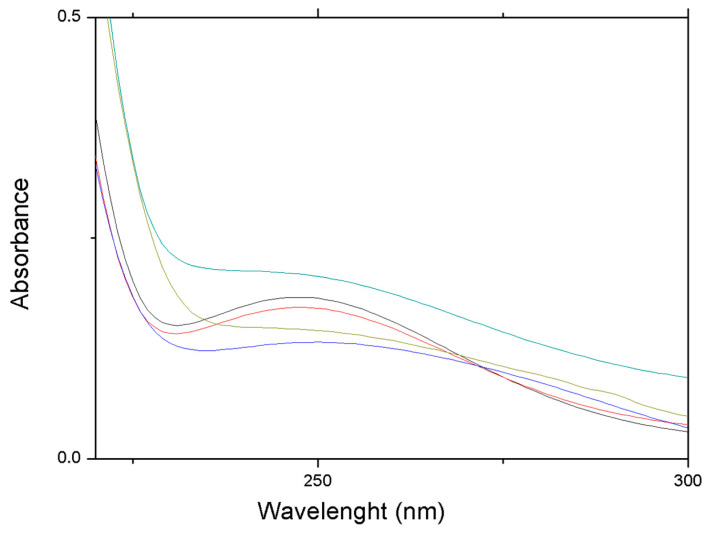
UV-Vis absorption spectra of supernatant solutions of NaAlg PMS (4 mg/mL) for increasing incubation times in PBS (pH = 7.4). Black: 15 min; red: 1 h; blue: 2 h; emerald: 24 h; olive:116 h.

**Figure 7 nanomaterials-13-02709-f007:**
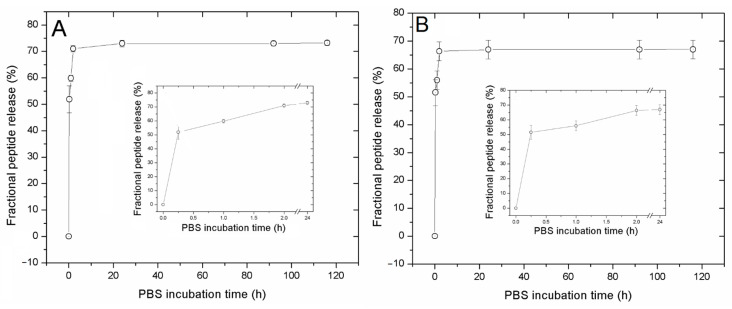
Cumulative fractional release of CIGB814 (%) from NaAlg PMS at different times (h) from fluorescence measurements (λ_em_ = 307 nm). (**A**) 1:8 CIGB814/NaAlgPMS formulation; (**B**) 1:16 CIGB814/NaAlgPMS formulation. Inset: cumulative fractional release of CIGB814 (%) during the first 2 h.

**Figure 8 nanomaterials-13-02709-f008:**
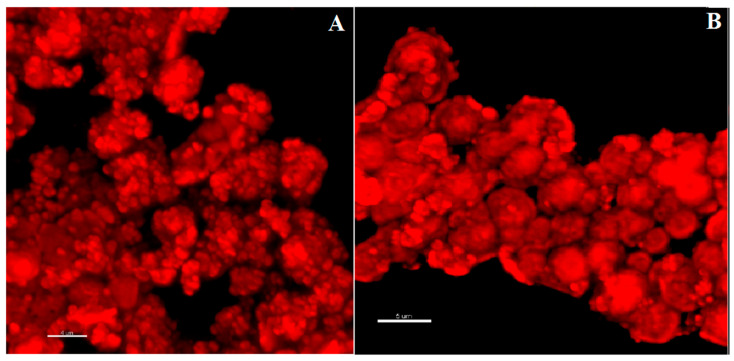
Three-dimensional CLSFM images of CIGB814/NaAlg(RBITC) microsponges. (**A**) Scale bar: 4 μm; (**B**) Scale bar: 5 μm.

**Figure 9 nanomaterials-13-02709-f009:**
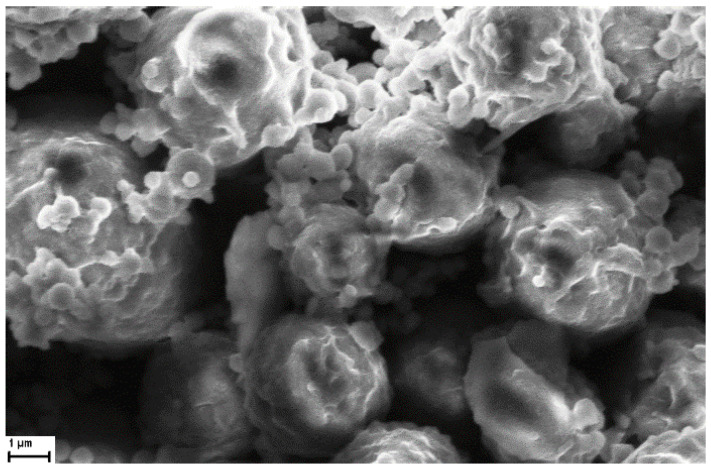
SEM image of CIGB814/NaAlgPMS (scale bar: 1 μm).

**Table 1 nanomaterials-13-02709-t001:** Inputs (Xs)—Selected processing parameters for the PMS diameter in Equation (6) (DoE).

	Parameter	Label	Unit	Low Level (−1)	High Level (+1)
*X* _1_	Cys amount	Cys	mg	20	30
*X* _2_	CDI amount	CDI	mg	20	30
*X* _3_	H_2_O amount	H_2_O	mL	200	300

## Data Availability

The data presented in this study are available on request from the corresponding author.

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
