# Peer review of "Alginate Microsponges as a Scaffold for Delivery of a Therapeutic Peptide against Rheumatoid Arthritis"

_nanomaterials, 2023, doi:10.3390/nano13192709_

Round 1

Reviewer 1 Report

The manuscript by Ariaudo et al. described the preparation and exploration of chemically crosslinked porous alginate particles for subsequent loading (by soaking) with the peptide CIGB814 (by soaking), which might be a candidate to treat rheumatoid arthritis. In principle, the paper is well written in terms of style and language. Still, in several parts of the manuscript, substantial improvements may be possible to increase clarity. Overall, the results are matching expectations. In several occasions the data were affected by experimental disturbance according to the honest statements of the authors and thus it should be critically evaluated if they are still suitable for publication. Furthermore, as there was basically no control of drug release (only burst), the applicability of the carrier system should be evaluated more critically.

The authors may consider the following comments:

1.       Line 18 and several other positions: IUPAC nomenclature should be used. Please critically check if your materials are truly “nanoporous” according to IUPAC definition.

2.       Lines 43-45: Microsponges are a formerly patented technology for porous particles made by emulsion polymerization. This is also how the authors have introduced them with reference [7]. As the term microsponge may currently be used in a broader sense (and also the authors did not use emulsion polymerization), it would be beneficial to describe their characteristic features in distinction to those of other porous microparticles in more detail.

3.       Lines 92-100: It is not clear why the authors emphasize the aggregation characteristics of their peptide. The phrase “as well as in their therapeutic formulation” sets doubts, whether or not this aggregation is desired and will be reversible.

4.       Fig. 1: Please increase the size of the structures. The labels should be in English (“minuti”). Please check for correct position of the brackets in the alginate structure above the arrow in panel B. Panal B also suggests, that the sodium salt (deprotonated carboxylic acid) is reacting to form an amide, which should be reconsidered.

5.       Lines 107-109. Purification of crosslinker and confirmation of structure should be described. Volume of HCl, duration of reaction, and termination step must be mentioned.

6.       Line 113: Information on crosslinker concentration is missing.

7.       Lines 116-117: missing word -> determined BY functionalizing

8.       Line 185: “PMS nanoporosity is determined by …” It would be meaningful to experimentally determine the overall porosity and pore size distribution of the microsponges. In fact, how can the authors be sure that they obtained porous systems rather than standard hydrogel particles?

9.       Line 187: Dendrimers are highly defined structures. The reviewer doubts that this term is applicable in its correct meaning to the structures reported here.

10.   Fig 2, Lines 207-211: Given the limitation of resolutions of optical microscopy, it may be hard to properly conclude on sizes like 0.6 µm from optical microscopy. It is furthermore hard to follow the authors’ statement that the particles have a dense inner phase from these images. The reviewer suggests to be very careful with conclusions on particle morphology based on these images. It may make sense to report the Electron microscopy data (Fig 9) along with Figure 2.

11.   Line 215: DLS results (particle mean diameters PDI) should be reported in a display item, considering the results of different formulations. The mean size data in Supp. Tab 3 suggest that mean sizes are at the upper edge of the measuring range of the Zetasizer and thus may not be trusted, if not being of extremely narrow distributions. Did you observe monomodal or multimodal distributions?

12.   Lines 222-239: The description on DOE suggests that the authors did not isolate or purify there crosslinker and thus are not sure, how much of the crosslinker was actually in the reaction mixture. This fact (if correct), which may diminish the precise understanding of the reaction and repeatability of the work by others, should be clearly stated and justified in the manuscript.

Furthermore, as the quantities of CDI and Cys seem to be static according to the methods section, it is not clear why they are actually considered as “variables” in the DOE studies.

13.   Fig 3B: Please clearly indicate what is meant with fractional loading. It seems the authors mean encapsulation efficiency relative to the peptide amount added to the preformed alginate particles and NOT the % loading of peptide relative to the polymer mass.

14.   Line 257, lines 283-286: The authors state that “diffuse light contamination” heavily affected the analysis of peptide loading for some conditions. What is the reason for this diffuse light? If the supernatant would truly be purified from particles, there should not be diffuse light. If the particles are still in the system, the measurements are not be valid and may not be reported.

15.   Fig. 3, Figure 4: It is not described in the methods section, how the shifts in the UV spectrum or the decreasing intensities in the fluorescence spectrum have been processed to obtain quantitative loading data.

16.   Line 296: The authors mention a degradation (which means bond cleavage!?) by osmotic effects, which is not clear in terms of the underlying mechanism. Agents supporting the cleavage of disulfide bonds may be discussed, if the authors truly mean that such cleavage of the network structure has occurred.

17.   Fig. 5 is reported without interpretation. The authors only state in line 300-302 that the results may not be quantitative, again due to disturbance by diffuse light.

18.   Fig 6: In previous Fig. the authors show decreasing fluorescence intensity in the supernatenat to suggest that the peptide is loaded into the particles. In Fig 6A and C, again decreasing fluorescence intensities are shown, in this case to show peptide release. This may be confusing for readers. The reviewer assumes that the decreasing fluorescence intensities in Fig 6A and C are a consequence of the sampling scheme (always replacing 50 vol% of the release medium at each time point). If this is true, than the Florescence spectra in Fig 6A and C may be of limited meaning and may be removed from the manuscript.

19.   Lines 318-325, Fig 6B/D: The authors did not observe a controlled release, but basically a burst release of almost all drug within less than 3-5 hours. This release pattern may need to be discussed in terms of the anticipated application. Why at all should the peptide by encapsulated by the particles? What would be the benefit compared to a pure peptide injection?

20.   Line 339: You are referring to Fig 7B and 8B, is this correct? Fig 8B does not look similarly homogeneous as Fig 7B.

21.   Line 343-344. The authors stated an intense red color at the outer surface of the particles and conclude that this means the peptide would be predominantly located there. First, it is not clear from the images (where also multiple particles are aggregated) that indeed the coloration at the surface is generally more intense than at the center. Second, it may be meaningful to discuss these data in the context of the labelling methodology, specifically how due intensity is linked to peptide position. According to the methods section, RBITC is used to label the alginate (see line 116-117; Supp. Fig 1) and not brought in contact with the peptide-loaded particles.

22.   Fig. 9: Crosssectional images would be beneficial to show that the particles are porous internally. It is not uncommon to see similarly collapsed surface structures in the SEM for standard hydrogel particles after drying.

In lines lines 375-377, the authors conclude from Fig 9B an predominant location of peptide at the surface. An alternative interpretation of Fig 9B is that a charging of the sample in the SEM led to a poor contrast, which is why the surface structure cannot be interpreted.

23.   Lines 395-397: The reviewer disagrees that the reported data justify to conclude that this is a slow drug delivery system.

Author Response

Please, see attachment

Reviewer 2 Report

Please provide the justification for pick this particular peptide CIGB814 to be formulated in alginate microsponges. Peptides are not bioavailable when taken orally. Is there evidence that this formulation improves its bioavailability?

From Figure 6 B and D, it seems the peptide is adsorbed on the surface of the microsponges. There were not significant changes in drug release at 24 h vs the last data point.

Round 2

Reviewer 1 Report

The reviewer has focused on the changes performed based on the previous comments of this reviewer. Despite many changes were performed as suggested or acceptable argumentation was provided, there remain several points where the reviewer comments did not result in serious reconsiderations:

1.       IUPAC defines microporous (< 2 nm), mesoporous (2-50 nm) and macroporous (> 50 nm) structures. In contrast to the authors statements, they did not use proper IUPAC nomenclature.

2.       The revised reaction scheme still suggests that carboxylate groups (deprotonated) react with an imidazole, which seems incorrect. It should be the protonated carboxylic acid which is reactive.

3.       The authors could not convincingly explain how their materials, claimed as “nanoporous”, would distinguish from standard hydrogels that also show pores after freeze drying. Beyond that, the pore sizes for hydrogels visible in SEM are highly affected by the sample preparation technique and typically represent artefacts of ice crystal formation (from freeze drying) rather than the native molecular network structure of the gel as synthesized.

Author Response

Please, see attachment
